# A Formal Model for Semantic Computing Based on Generalized Probabilistic Automata

**DOI:** 10.3390/e21090903

**Published:** 2019-09-17

**Authors:** Guangjian Huang, Shahbaz Hassan Wasti, Lina Wei, Yuncheng Jiang

**Affiliations:** School of Computer Science, South China Normal University, Guangzhou 510631, China; 2017010160@m.scnu.edu.cn (G.H.); shahbazwasti@gmail.com (S.H.W.); 2018010188@m.scnu.edu.cn (L.W.)

**Keywords:** probabilistic automata, probability distribution, related concept, semantic computing

## Abstract

In most previous research, “semantic computing” refers to computational implementations of semantic reasoning. It lacks support from the formal theory of computation. To provide solid foundations for semantic computing, researchers propose a different understanding of semantic computing based on finite automata. This approach provides a computer theoretical approach to semantic computing. But finite automata are not capable enough to deal with imprecise knowledge. Therefore, in this paper, we provide foundations for semantic computing based on probabilistic automata. Even though traditional probabilistic automata can handle imprecise knowledge, their limitation resides in their being defined on a fixed finite input alphabet. This deeply restricts the abilities of automata. In this paper, we rebuild traditional probabilistic automata for semantic computing. Furthermore, our new probabilistic automata are robust enough to handle any alphabet as input. They have better performances in many applications. We provide an application for weather forecasting, a domain for which traditional probabilistic automata are not effective due to their finite input alphabet. Our new probabilistic automata can overcome these limitations.

## 1. Introduction

Semantic similarity is applied in plentiful applications about artificial intelligence and computational linguistics, such as word sense disambiguation, information retrieval, knowledge acquisition and natural language processing [1,2,3]. But most research on Semantic Computing (SC) are devoted to the following three fields: (1) formal models for theoretical foundations of SC, such as description logics, ontology reasoning and answer set programming [4,5]; (2) fundamental languages and technologies for interoperability and reuse of information including RDF, RDFS, the OWL family of languages, the WSML family of languages, and SPARQL [6,7]; (3) some applications of SC [8,9,10]. As a result, “semantic computing” refers to computational implementations of semantic reasoning (e.g., ontology reasoning, rule reasoning, semantic query and semantic search) but it is not from the perspective of the formal theory of computation. To provide solid foundations for semantic computing, it is significant to bridge the gap between semantic computing and the formal theory of computation.

Recently, Reference [11] proposed a different understanding of semantic computing based on finite automata. Unlike previous research, it explains semantic computing from the computation theory perspective. Their models for semantic computing are based on finite automata. However, finite automata cannot deal with imprecise knowledge, which is the advantage of probabilistic automata. Unfortunately, traditional probabilistic automaton $M=(Q,\Sigma ,\delta ,q_{0},F)$ is not capable enough, because it is defined on a finite input alphabet Σ. When an undefined input a∉Σ is transmitted to automaton in the applications, it will be invalid. But in the applications, the inputs transmitted from users are unpredictable. Users may input synonyms, equivalent concepts, acronyms or even some words with spelling errors. The fixed finite set of inputs restrict the abilities of automata deeply. For robustness, we prefer to build an automaton which can take any alphabet as input and semantic computing is the key to build such automata.

In this paper, we rebuild traditional probabilistic automata to provide formal models for semantic computing, based on semantic similarity of two symbols or concepts [2,3,12,13,14]. We will apply the existing methods in semantic similarity to probabilistic automata directly. For more information about semantic similarity, readers are encouraged to read our previous work [2,3,15].

The intuitive idea to rebuild a traditional probabilistic automaton is that we redefine the string S∉Σ as its corresponding most similar input string $S^{\prime }\in \Sigma$, by the methods of semantic similarity. Specifically, when automata need to deal with an undefined input string *S*, we find its similar string $S^{\prime }$ in the input alphabet. Then we handle string *S* in the way similar to $S^{\prime }$. As a result, probabilistic automata can take any string as input, whether defined or not. In the following section, we will propose an application of the weather forecast. Because the weather factor has been changing all the time, it is impossible to define all the weather factors in a finite input alphabet. Therefore, traditional probabilistic automata are not competent in the weather forecast. But with the methods in semantic computing, for any undefined input, our new models can take them as corresponding most similar defined inputs. Moreover, our new models provide a computer theoretical approach to semantic computing.

The rest of this paper is organized as follows. In Section 2, we briefly recall some basic notions of probabilistic automata. In Section 3, we build the models for semantic computing based on probabilistic automata by semantic similarity. Section 4 gives an application for the weather forecast, and Section 5 concludes this paper.

## 2. Related Work

For convenience, in this section, we will briefly recall some basic notions of probabilistic automata. There are more details in References [16,17,18,19,20].

**Definition** **1**(Probability distribution)**.** *Suppose U is a finite set. A probability distribution on U is a function f from U to [0,1], if $\sum_{x\in U}f\left(x\right)=1$. The set {x∈U|f(x)>0} is called the support of U. The set of all probability distributions on U is denoted as D(U). A probability distribution f∈D(U) is denoted as follows [21]:*
$$f=f\left(x_{1}\right)/x_{1}+f\left(x_{2}\right)/x_{2}+f\left(x_{3}\right)/x_{3}\cdots .$$
*For any λ∈[0,1],f∈D(U), scalar multiplication λ·f:U→[0,1] is defined as (λ·f)(x)=λ·f(x).*

**Definition** **2**(PA)**.** *A probabilistic automaton (PA) is a five-tuple $M=(Q,\Sigma ,\delta ,q_{0},F)$, where*
Q is a finite set of states,Σ *is a finite input alphabet,*δ:Q×Σ→D(Q) is the transition function,$q_{0}\in Q$ is the start state, $q_{0}=\sum_{i}\alpha_{i}q_{i}$ with $\sum \alpha_{i}=1,a_{i}\in \left[0,1\right]$,F⊆Q is the set of accept states.

**Remark** **1.**
*δ(q,a)(p) denotes the probability of automaton entering state p with input a from state q. In [17], transition function is defined in an equivalent form $\delta^{\prime }:Q\times \Sigma \times Q\to \left[0,1\right]$ satisfying $\sum_{p\in Q}\delta^{\prime }\left(q,a,p\right)=1$. $\delta^{\prime }\left(q,a,p\right)$ presents the probability of automaton entering state p with input a from state q, i.e., $\delta \left(q,a\right)\left(p\right)=\delta^{\prime }\left(q,a,p\right)$. In the form δ:Q×Σ→D(Q), we do not have a constraint similar to $\sum_{p\in Q}\delta^{\prime }\left(q,a,p\right)=1$, because by complete probability formula we can get that for any f∈D(U),∑f(x)=1. For convenience, both forms of transition function are denoted as δ, i.e., δ(q,a)(p)=δ(q,a,p).*


**Definition** **3**(Semantic similarity)**.** *Let U be a set of concepts or symbols. The semantic similarity of two concepts a,b∈U is defined as sim:U×U→I, where I is unit interval.*

In traditional approaches, Reference [1] computes semantic similarity of concepts based on WordNet. Our previous work [2,3,15] is based on Wikipedia. Couto, F.M. et al. [22] proposed an approach by ontology. In this paper, we combine the theory of semantic similarity and computation to construct a new model of probabilistic automaton.

## 3. Formal Model of Semantic Computing Based on Probabilistic Automata

In this section, we present formal models for semantic computing by generalizing the automata mentioned in the above section. Firstly, we consider a simple case, extending automata by equivalent concepts. Then we investigate a general situation, generalizing automata by semantically related concepts.

### 3.1. Probabilistic Automata under Equivalent Concepts

Suppose that $M=(Q,\Sigma ,\delta ,q_{0},F)$ is a PA. One of the limitation of a PA *M* is that *M* is restricted by finite input alphabet Σ. Suppose $x=a_{1}a_{2}\cdots a_{n}\in \Sigma^{*}$ ($\Sigma^{*}$ is the set of all strings over Σ, containing empty string ε) and $x^{\prime }=a_{1}^{\prime }a_{2}^{\prime }\cdots a_{n}^{\prime }\notin \Sigma^{*}$. For any 1≤i≤n, $a_{i}$ and $a_{i}^{\prime }$ are equivalent concept or synonym in semantics. For instance, $x=a_{1}a_{2}\in \Sigma^{*}$ and $x^{\prime }=a_{1}^{\prime }a_{2}^{\prime }\notin \Sigma^{*}$ where $a_{1}=true,a_{2}=valid$ and $a_{1}^{\prime }=correct,a_{2}^{\prime }=authorized$. *M* is valid with input *x*, but not $x^{\prime }$. But the question is that in many applications, the inputs transmitted from users are unpredictable. Previously, we defined $a_{1}=true$ as a legal input. But in the application, users may input $a_{1}^{\prime }=correct$ instead of $a_{1}=true$. Because $a_{1}^{\prime }\notin \Sigma$, automaton *M* is invalid with input $a_{1}^{\prime }$. For robustness, the automaton *M* is supposed to have the ability to take equivalent concepts or synonyms as inputs as well.

**Definition** **4.**
*Suppose *Σ* is an alphabet and π is the alphabet of all the possible symbols. For any b∈Σ,a∈π, if sim(a,b)=1, then b is called an equivalent concept or synonym of a in *Σ*, denoted as $a^{e}=b$. If a∈Σ then $a^{e}=a$. If there does not exist any equivalent concept or synonym in *Σ*, then $a^{e}=\varepsilon$ where ε is empty string. Therefore, for any a∈π, there exist $a^{e}\in \Sigma^{*}$. Both equivalent concepts and synonyms are took as equivalent concepts in this paper.*


In the case of $a^{e}=\varepsilon$, the transition function δ:Q×Σ→D(Q) of a PA *M* need to be generalized to a function $Q\times \Sigma^{*}\to D\left(Q\right)$ and for convenience, we still denote it as δ. For q∈Q,
$$\delta \left(q,A\right)=\left\{\begin{array}{c} \delta (q,\varepsilon )=1/q,\;\;if\;\;A=\varepsilon ,\\ \delta \left(q,a_{1}a_{2}\right)=\sum_{p\in Q}\delta \left(q,a_{1}\right)\left(p\right)\cdot \delta \left(p,a_{2}\right),\;\;if\;\;A=a_{1}a_{2}\;\;\mathrm{and}\;\;a_{1}\in \Sigma^{*},a_{2}\in \Sigma . \end{array}\right.$$

The language accepted by PA *M* is defined as a function $L_{M}:\Sigma^{*}\to \left[0,1\right]$: for any $x=a_{1}a_{2}\cdots a_{n}\in \Sigma^{*}$,
$$L_{M}\left(x\right)=\sum_{q\in F,q_{i}\in Q}\left\{\delta \left(q_{0},a_{1},q_{1}\right)\cdot \delta \left(q_{1},a_{2},q_{2}\right)\cdots \delta \left(q_{n-1},a_{n},q\right)\right\},i=1,2\cdots n-1,$$
where $\delta \left(q_{0},a_{1},q_{1}\right)=\sum_{i}\alpha_{i}\delta \left(q_{i},a_{1},q_{1}\right)$, if $q_{0}=\sum_{i}\alpha_{i}q_{i}$.

**Definition** **5**(DPEC)**.** *A deterministic probabilistic automaton for semantic computing under equivalent concepts (DPEC) is a seven-tuple $M=(Q,\Sigma ,\delta ,\Sigma^{\prime },\delta^{\prime },q_{0},F)$, where*
Q is a finite set of states,Σ *is a finite input alphabet,*δ:Q×Σ→D(Q) is a internal transition function,$q_{0}\in Q$ is the start state, $q_{0}=\sum_{i}\alpha_{i}q_{i}$ with $\sum \alpha_{i}=1,a_{i}\in \left[0,1\right]$,F⊆Q is the set of accept states,$\Sigma^{\prime }=\left\{a|\exists !a^{e}\in \Sigma ,a\in \pi \right\}$, where $\exists !a^{e}\in \Sigma$ means there exist only one $a^{e}\in \Sigma$ for every a,*$\delta^{\prime }:Q\times \left(\Sigma \cup \Sigma^{\prime }\right)\to D\left(Q\right)$ is a generalized transition function of δ:*$$\delta^{\prime }\left(q,a\right)=\left\{\begin{array}{c} \delta (q,a),\;\;if\;\;a\in \Sigma ,\\ \delta \left(q,a^{e}\right),\;\;if\;\;a\in \Sigma^{\prime }. \end{array}\right.$$

In the traditional perspective, $\Sigma \cup \Sigma^{\prime }$ is the alphabet of *M* and $\delta^{\prime }$ is the transition function of *M*. But in this paper, the transition function δ is called internal transition function and $\delta^{\prime }$ is called generalized transition function. Simply, by transition function of *M*, we mean $\delta^{\prime }$. An input in Σ is called an original input and an input in $\Sigma^{\prime }$ is called a generalized input.

**Example** **1**(Command order)**.** *For instance, we consider a sample case of command order. We consider a simple case that every time users input a command, automaton will enter a state based on probability distribution. Suppose that in PA $M=(Q,\Sigma ,\delta ,q_{0},F)$, Q={q,p}. Σ={true,false}. $q_{0}=q,F=Q$. δ is defined as:*
$$\begin{array}{c} \delta (q,true,q)=0.7,\delta (q,false,q)=0.8,\\ \delta (q,true,p)=0.3,\delta (q,false,p)=0.2,\\ \delta (p,true,q)=0.1,\delta (p,false,q)=0.4,\\ \delta (p,true,p)=0.9,\delta (p,false,p)=0.6. \end{array}$$
*In PA M, “true” and “false” in *Σ* are commands input by users. δ(q,true,p)=0.3 means when users are in state q and input command “true”, the probability of that he will enter state p is 0.3. Others are similar. But in application, users may input “right” and “wrong” instead of “true” and “false”, because they are equivalent concepts for users. In our daily life, there are many equivalent concepts and users cannot know which are legal commands.*
*In order to empower M with the ability of taking equivalent concepts as commands, we rebuild PA M as a DPEC $M^{\prime }=\left(Q,\Sigma ,\delta ,\Sigma^{\prime },\delta^{\prime },q_{0},F\right)$ as follows. Suppose that we need to consider the additional case that users will input* “wrong” *and* “right”*, i.e., $\Sigma^{\prime }=\left\{wrong,right\right\}$. For other cases, it is analogous. Define $\delta^{\prime }\left(k,a,k^{\prime }\right)=\delta \left(k,a,k^{\prime }\right)$, for $k,k^{\prime }\in Q,a\in \Sigma$. Suppose*
sim(right,true)=1,sim(wrong,false)=1,
*and for other cases $sim\left(a,a^{\prime }\right)=0,a^{\prime }\in \Sigma^{\prime },a\in \Sigma$. For $a\in \Sigma^{\prime }$, $\delta^{\prime }$ is defined as:*
$$\begin{array}{c} \delta^{\prime }\left(q,right,p\right)=\delta \left(q,true,p\right)=0.3,\\ \delta^{\prime }\left(q,wrong,p\right)=\delta \left(q,false,p\right)=0.2,\\ \delta^{\prime }\left(q,right,q\right)=\delta \left(q,true,q\right)=0.7,\\ \delta^{\prime }\left(q,wrong,q\right)=\delta \left(q,false,q\right)=0.8,\\ \delta^{\prime }\left(p,right,q\right)=\delta \left(p,true,q\right)=0.1,\\ \delta^{\prime }\left(p,wrong,q\right)=\delta \left(p,false,q\right)=0.4,\\ \delta^{\prime }\left(p,right,p\right)=\delta \left(p,true,p\right)=0.9,\\ \delta^{\prime }\left(p,wrong,p\right)=\delta \left(p,false,p\right)=0.6. \end{array}$$
*With the above extended $\Sigma^{\prime }$ and $\delta^{\prime }$, when we suppose to get “true” from users, but users input “right”, PA M is invalid. However, DPEC $M^{\prime }$ is still valid. Similarly, we can deal with spelling error, such as “ture”. The key to do this is the semantic similarity function sim. If we define sim(true,ture)=1 and $\delta^{\prime }\left(k,ture,k^{\prime }\right)=\delta^{\prime }\left(k,true,k^{\prime }\right),k,k^{\prime }\in Q$, then spelling error “ture” can be taken as a legal input.*


But in many cases, for some a∈π, there may exist several $a^{e}\in \Sigma$. This is defined as the following nondeterministic case.

**Definition** **6**(NPEC)**.** *A nondeterministic probabilistic automaton for semantic computing under equivalent concepts (NPEC) is a seven-tuple $M=(Q,\Sigma ,\delta ,\Sigma^{\prime },\delta^{\prime },q_{0},F)$, where*
$Q,\Sigma ,\delta ,q_{0},F$ are the same in DPEC,$\Sigma^{\prime }=\left\{a|a\in \pi ,\exists a^{e}\in \Sigma \right\}$,*$\delta^{\prime }:Q\times \left(\Sigma \cup \Sigma^{\prime }\right)\to D\left(Q\right)$ is a generalized transition function of δ:*$$\delta^{\prime }\left(q,a\right)=\left\{\begin{array}{c} \delta (q,a),\;\;if\;\;a\in \Sigma ,\\ \sum_{a^{e}\in \Sigma }\delta \left(q,a^{e}\right),\;\;if\;\;\sum_{a^{e}\in \Sigma }\delta \left(q,a^{e}\right)\leq 1,a\in \Sigma^{\prime },\\ 1,\;\;if\;\;\sum_{a^{e}\in \Sigma }\delta \left(q,a^{e}\right)>1,a\in \Sigma^{\prime }. \end{array}\right.$$

In the traditional perspective, $\Sigma \cup \Sigma^{\prime }$ is the alphabet of NPEC *M* and $\delta^{\prime }$ is the transition function of NPEC *M*.

**Example** **2**(Command order)**.** *Suppose that the designers of the PA M in above Example 1 get the feedback that users get an error report with inputs “right” and “wrong” frequently. In order to make it more convenient for users, the designers improve it as follows. In PA $M=(Q,\Sigma ,\delta ,q_{0},F)$, *Σ* is redefined as Σ={true,false,right,wrong}. δ is redefined as:*
$$\begin{array}{c} \delta (q,right,p)=\delta (q,true,p)=0.3,\\ \delta (q,wrong,p)=\delta (q,false,p)=0.2,\\ \delta (q,right,q)=\delta (q,true,q)=0.7,\\ \delta (q,wrong,q)=\delta (q,false,q)=0.8,\\ \delta (p,right,q)=\delta (p,true,q)=0.1,\\ \delta (p,wrong,q)=\delta (p,false,q)=0.4,\\ \delta (p,right,p)=\delta (p,true,p)=0.9,\\ \delta (p,wrong,p)=\delta (p,false,p)=0.6. \end{array}$$
*But there are so many equivalent concepts. What if some users may input “correct” instead of “right” or “true”?*

*In this case, we rebuild PA M as a NPEC $M^{\prime }=\left(Q,\Sigma ,\delta ,\Sigma^{\prime },\delta^{\prime },q_{0},F\right)$ as follows. In this case, $\Sigma^{\prime }=\left\{correct\right\}$. Define $\delta^{\prime }\left(k,a,k^{\prime }\right)=\delta \left(k,a,k^{\prime }\right)$, for $k,k^{\prime }\in Q,a\in \Sigma$. Because δ(q,right,p)=δ(q,true,p)=0.3 and “correct” is equivalent to “right” and “true”, so we define $\delta^{\prime }\left(q,correct,p\right)=\delta \left(q,right,p\right)+\delta \left(q,true,p\right)=0.3+0.3=0.6$. Analogously, $\delta^{\prime }$ can be defined as:*
$$\begin{array}{c} \delta^{\prime }\left(q,right,p\right)=\delta \left(q,right,p\right)+\delta \left(q,true,p\right)=0.6,\\ \delta^{\prime }\left(q,right,q\right)=min\left\{\delta \left(q,right,q\right)+\delta \left(q,true,q\right),1\right\}=1,\\ \delta^{\prime }\left(p,right,q\right)=\delta \left(p,right,q\right)+\delta \left(p,true,q\right)=0.2,\\ \delta^{\prime }\left(p,right,p\right)=min\left\{\delta \left(p,right,q\right)+\delta \left(p,true,p\right),1\right\}=1. \end{array}$$

*With the above extended $\Sigma^{\prime }$ and $\delta^{\prime }$, when we suppose to get “true” and “right” from users, but users input “correct”, PA M is invalid. However, NPEC $M^{\prime }$ is still valid. Similarly, we can deal with other equivalent concepts.*


The key point is that the inputs transmitted from users are unpredictable. The finite input alphabet Σ in PA *M* cannot take every case into account. But NPEC is competent. Based on the research on semantic computing introduced previously, we can apply their methods of semantic computing to PA *M* directly. Once an undefined input is transmitted to *M*, we find its equivalent inputs and modify $\delta^{\prime }$, then it can be taken as a legal input. But most of the time, we can only find similar inputs instead of equivalent concepts. We leave this case to the next section.

The transition function of a DPEC or NPEC $M^{\prime }$ can be generalized to a function $\delta^{\prime }:Q\times {\left(\Sigma \cup \Sigma^{\prime }\right)}^{*}\to D\left(Q\right)$ and for convenience, we still denote it as $\delta^{\prime }$:$$\delta^{\prime }\left(q,A\right)=\left\{\begin{array}{c} \delta^{\prime }\left(q,\varepsilon \right)=1/q,\;\;if\;\;A=\varepsilon ,q\in Q,\\ \delta^{\prime }\left(q,a_{1}a_{2}\right)=\sum_{p\in Q}\delta^{\prime }\left(q,a_{1}\right)\left(p\right)\cdot \delta^{\prime }\left(p,a_{2}\right),\;\;if\;\;A=a_{1}a_{2}\;\;and\;\;a_{1}\in {\left(\Sigma \cup \Sigma^{\prime }\right)}^{*},\\ a_{2}\in \Sigma \cup \Sigma^{\prime },q\in Q. \end{array}\right.$$

By the above generalized transition function, we can get the robustness of a DPEC or NPEC $M^{\prime }$:

**Theorem** **1**(Robustness)**.** *Suppose that $M^{\prime }$ is a DPEC or NPEC, then $M^{\prime }$ can take any string of symbols $A\in \pi^{*}$ as an input, i.e., the transition function of a DPEC or NPEC $M^{\prime }$ can be generalized to a function $\delta^{\prime }:Q\times \pi^{*}\to D\left(Q\right)$.*

**Proof.** For any q∈Q, if A=ε, then $\delta^{\prime }\left(q,A\right)=\delta^{\prime }\left(q,\varepsilon \right)=1/q$. If A≠ε then suppose $A=a_{1}a_{2}\cdots a_{n}$ where $a_{i}\in \pi ,1\leq i\leq n$. If $a_{i}=\varepsilon$, then $\delta^{\prime }\left(q,a_{i}\right)=\delta^{\prime }\left(q,\varepsilon \right)=1/q$. If $a_{i}\in \Sigma$, then $\delta^{\prime }\left(q,a_{i}\right)=\delta \left(q,a_{i}\right)$. If $a_{i}\notin \Sigma$, then $\delta^{\prime }\left(q,a_{i}\right)=\delta \left(q,a_{i}^{e}\right)$. Recall that if a∈Σ, then $a^{e}=a$. Therefore
$$\begin{array}{c} \delta^{\prime }\left(q,A\right)\\ =\delta^{\prime }\left(q,a_{1}a_{2}\cdots a_{n}\right)\\ =\sum_{p_{1}p_{2}\cdots p_{n-1}\in Q}\delta^{\prime }\left(q,a_{1}\right)\left(p_{1}\right)\cdot \delta^{\prime }\left(p_{1},a_{2}\right)\left(p_{2}\right)\cdots \delta^{\prime }\left(p_{n-1},a_{n}\right)\\ =\sum_{p_{1}p_{2}\cdots p_{n-1}\in Q}\delta \left(q,a_{1}^{e}\right)\left(p_{1}\right)\cdot \delta \left(p_{1},a_{2}^{e}\right)\left(p_{2}\right)\cdots \delta \left(p_{n-1},a_{n}^{e}\right), \end{array}$$
which means $M^{\prime }$ can take any string of symbols $A\in \pi^{*}$ as an input, i.e., the transition function of $M^{\prime }$ can be generalized to a function $\delta^{\prime }:Q\times \pi^{*}\to D\left(Q\right)$. □

As analysed above, in many applications of automata, the inputs transmitted from users are unpredictable. If an automaton can take any string of symbols as an input, it can be applied to more fields. The robustness of probabilistic automata empowers it to be more stable. Intuitively, a DPEC or NPEC is a semantic expansion of PA which releases PA from a finite input alphabet.

**Definition** **7.**
*The language accepted by a DPEC or NPEC $M^{\prime }$ is defined as a function $L_{M^{\prime }}:\pi^{*}\to \left[0,1\right]$: for any $x=a_{1}a_{2}\cdots a_{n}\in \pi^{*}$,*
$$L_{M^{\prime }}\left(x\right)=\sum_{q\in F,q_{i}\in Q}\left\{\delta^{\prime }\left(q_{0},a_{1},q_{1}\right)\cdot \delta^{\prime }\left(q_{1},a_{2},q_{2}\right)\cdots \delta^{\prime }\left(q_{n-1},a_{n},q\right)\right\},i=1,2\cdots n-1.$$


DPEC and NPEC are semantic generalizations of PA for semantic computing. To show this relationship, we can define DPEC and NPEC in the following equivalent forms.

**Definition** **8**(DPEC)**.** *A deterministic probabilistic automaton for semantic computing under equivalent concepts (DPEC) is a seven-tuple $M^{\prime }=\left(Q,\Sigma ,\delta ,\Sigma^{\prime },\delta^{\prime },q_{0},F\right)$, where*
$M=(Q,\Sigma ,\delta ,q_{0},F)$ is a PA,$\Sigma^{\prime }=\left\{a|a\in \pi ,\exists !a^{e}\in \Sigma \right\}$,*$\delta^{\prime }:Q\times {\left(\Sigma \cup \Sigma^{\prime }\right)}^{*}\to D\left(Q\right)$ is a generalized transition function of δ:*$$\delta^{\prime }\left(q,a\right)=\left\{\begin{array}{c} \delta (q,a),\;\;if\;\;a\in \Sigma ,\\ \delta \left(q,a^{e}\right),\;\;if\;\;a\in \Sigma^{\prime }. \end{array}\right.$$

**Definition** **9**(NPEC)**.** *A nondeterministic probabilistic automaton for semantic computing under equivalent concepts (NPEC) is a seven-tuple $M^{\prime }=\left(Q,\Sigma ,\delta ,\Sigma^{\prime },\delta^{\prime },q_{0},F\right)$, where*
$M=(Q,\Sigma ,\delta ,q_{0},F)$ is a PA,$\Sigma^{\prime }=\left\{a|\exists a^{e}\in \Sigma ,a\in \pi \right\}$,*$\delta^{\prime }:Q\times {\left(\Sigma \cup \Sigma^{\prime }\right)}^{*}\to D\left(Q\right)$ is a generalized transition function of δ:*$$\delta^{\prime }\left(q,a\right)=\left\{\begin{array}{c} \delta (q,a),\;\;if\;\;a\in \Sigma ,\\ \sum_{a^{e}\in \Sigma }\delta \left(q,a^{e}\right),\;\;if\;\;\sum_{a^{e}\in \Sigma }\delta \left(q,a^{e}\right)\leq 1,a\in \Sigma^{\prime },\\ 1,\;\;if\;\;\sum_{a^{e}\in \Sigma }\delta \left(q,a^{e}\right)>1,a\in \Sigma^{\prime }. \end{array}\right.$$

The language accepted by DPEC (or NPEC) and PA have the following property.

**Theorem** **2**(Semantic generalization)**.** *Suppose that DPEC $M^{\prime }=\left(Q,\Sigma ,\delta ,\Sigma^{\prime },\delta^{\prime },q_{0},F\right)$ is a semantic generalizationof PA $M=(Q,\Sigma ,\delta ,q_{0},F)$ and the language accepted by them are function $L_{M^{\prime }}$ and $L_{M}$ respectively. The languages accepted by them have the following properties:*
*1*.for any $x\in \Sigma^{*}$, $L_{M^{\prime }}\left(x\right)=L_{M}\left(x\right)$,*2*.for any $x\in \pi^{*}$, there exist $x^{\prime }\in \Sigma^{*}$ such that $L_{M^{\prime }}\left(x\right)=L_{M^{\prime }}\left(x^{\prime }\right)=L_{M}\left(x^{\prime }\right)$.

**Proof.** For DPEC $M^{\prime }$ and PA *M*, if x=ε, then $\delta^{\prime }\left(q,x\right)=\delta \left(q,x\right)=1/q$, for any q∈Q. Obviously, $L_{M^{\prime }}\left(x\right)=L_{M}\left(x\right)$. If x≠ε, (1) for any $x=a_{1}a_{2}\cdots a_{n}\in \Sigma^{*},a_{i}\in \Sigma ,1\leq i\leq n-1$,
$$L_{M}\left(x\right)=\sum_{q\in F,q_{i}\in Q}\left\{\delta \left(q_{0},a_{1},q_{1}\right)\cdot \delta \left(q_{1},a_{2},q_{2}\right)\cdots \delta \left(q_{n-1},a_{n},q\right)\right\}.$$Since $\delta \left(q,a,p\right)=\delta^{\prime }\left(q,a,p\right)$, for any a∈Σ,q,p∈Q, we get
$$\begin{array}{c} L_{M^{\prime }}\left(x\right)\\ =\sum_{q\in F,q_{i}\in Q}\left\{\delta^{\prime }\left(q_{0},a_{1},q_{1}\right)\cdot \delta^{\prime }\left(q_{1},a_{2},q_{2}\right)\cdots \delta^{\prime }\left(q_{n-1},a_{n},q\right)\right\}\\ =\sum_{q\in F,q_{i}\in Q}\left\{\delta \left(q_{0},a_{1},q_{1}\right)\cdot \delta \left(q_{1},a_{2},q_{2}\right)\cdots \delta \left(q_{n-1},a_{n},q\right)\right\}\\ =L_{M}\left(x\right). \end{array}$$(2) For any $x=a_{1}a_{2}\cdots a_{n}\in \pi^{*},a_{i}\in \pi ,1\leq i\leq n-1$,
$$L_{M^{\prime }}\left(x\right)=\sum_{q\in F,q_{i}\in Q}\left\{\delta^{\prime }\left(q_{0},a_{1},q_{1}\right)\cdot \delta^{\prime }\left(q_{1},a_{2},q_{2}\right)\cdots \delta^{\prime }\left(q_{n-1},a_{n},q\right)\right\}.$$For $a\in \Sigma^{\prime }$, $\delta^{\prime }\left(q,a\right)=\delta \left(q,a^{e}\right)$ and recall that $a^{e}=a$ if a∈Σ. Let $x^{\prime }=a_{1}^{e}a_{2}^{e}\cdots a_{n}^{e}$, then $x^{\prime }\in \Sigma^{*}$.
$$\begin{array}{c} L_{M^{\prime }}\left(x\right)\\ =\sum_{q\in F,q_{i}\in Q}\left\{\delta^{\prime }\left(q_{0},a_{1},q_{1}\right)\cdot \delta^{\prime }\left(q_{1},a_{2},q_{2}\right)\cdots \delta^{\prime }\left(q_{n-1},a_{n},q\right)\right\}\\ =\sum_{q\in F,q_{i}\in Q}\left\{\delta \left(q_{0},a_{1}^{e},q_{1}\right)\cdot \delta \left(q_{1},a_{2}^{e},q_{2}\right)\cdots \delta \left(q_{n-1},a_{n}^{e},q\right)\right\}\\ =L_{M^{\prime }}\left(x^{\prime }\right). \end{array}$$Since $x^{\prime }\in \Sigma^{*}$, by (1) we can get $L_{M^{\prime }}\left(x^{\prime }\right)=L_{M}\left(x^{\prime }\right)$. Therefore, $L_{M^{\prime }}\left(x\right)=L_{M^{\prime }}\left(x^{\prime }\right)=L_{M}\left(x^{\prime }\right)$. □

**Theorem** **3**(Semantic generalization)**.** *Suppose that NPEC $M^{\prime }=\left(Q,\Sigma ,\delta ,\Sigma^{\prime },\delta^{\prime },q_{0},F\right)$ is a semantic generalization of PA $M=(Q,\Sigma ,\delta ,q_{0},F)$ and the language accepted by them are function $L_{M^{\prime }}$ and $L_{M}$ respectively. The languages accepted by them have the following properties:*
*1*.for any $x\in \Sigma^{*}$, $L_{M^{\prime }}\left(x\right)=L_{M}\left(x\right)$,*2*.for any $x\in \pi^{*}$, there exist $x^{\prime }\in \Sigma^{*}$ such that $L_{M^{\prime }}\left(x\right)\geq L_{M^{\prime }}\left(x^{\prime }\right)=L_{M}\left(x^{\prime }\right)$.

**Proof.** For NPEC $M^{\prime }$ and PA *M*, if x=ε or $x=a_{1}a_{2}\cdots a_{n}\in \Sigma^{*},a_{i}\in \Sigma ,1\leq i\leq n$, it is similar to DPEC.For any $x=a_{1}a_{2}\cdots a_{n}\in \pi ,a_{i}\in \pi ,1\leq i\leq n-1$,
$$L_{M^{\prime }}\left(x\right)=\sum_{q\in F,q_{i}\in Q}\left\{\delta^{\prime }\left(q_{0},a_{1},q_{1}\right)\cdot \delta^{\prime }\left(q_{1},a_{2},q_{2}\right)\cdots \delta^{\prime }\left(q_{n-1},a_{n},q\right)\right\},i=1,2\cdots n-1.$$For $a\in \Sigma^{\prime }$, $\delta \left(q,a^{e},p\right)\leq \sum_{a^{e}\in \Sigma }\delta \left(q,a^{e},p\right)$ and $\delta (q,a^{e},p)\leq 1$, therefore,
$$\delta^{\prime }\left(q,a,p\right)=min\left\{\sum_{a^{e}\in \Sigma }\delta \left(q,a^{e},p\right),1\right\}\geq \delta \left(q,a^{e},p\right).$$Recall that $a^{e}=a$ if a∈Σ. Let $x^{\prime }=a_{1}^{e}a_{2}^{e}\cdots a_{n}^{e}$, then $x^{\prime }\in \Sigma^{*}$.
$$\begin{array}{c} L_{M^{\prime }}\left(x\right)\\ =\sum_{q\in F,q_{i}\in Q}\left\{\delta^{\prime }\left(q_{0},a_{1},q_{1}\right)\cdot \delta^{\prime }\left(q_{1},a_{2},q_{2}\right)\cdots \delta^{\prime }\left(q_{n-1},a_{n},q\right)\right\}\\ \geq \sum_{q\in F,q_{i}\in Q}\left\{\delta \left(q_{0},a_{1}^{e},q_{1}\right)\cdot \delta \left(q_{1},a_{2}^{e},q_{2}\right)\cdots \delta \left(q_{n-1},a_{n}^{e},q\right)\right\}\\ =L_{M^{\prime }}\left(x^{\prime }\right). \end{array}$$Since $x^{\prime }\in \Sigma^{*}$, by (1) we can get $L_{M^{\prime }}\left(x^{\prime }\right)=L_{M}\left(x^{\prime }\right)$. Therefore, $L_{M^{\prime }}\left(x\right)\geq L_{M^{\prime }}\left(x^{\prime }\right)=L_{M}\left(x^{\prime }\right)$. □

The intuitive idea of the above two theorems is that because DPEC and NPEC $M^{\prime }$ are semantically generalized from PA *M*, then for an original input in Σ, they will get the same result. For a generalized input in $\Sigma^{\prime }$, if there is only one equivalent concept, they will get an equivalent result too. But if there are several equivalent concepts, the generalized input will include all the equivalent cases, then NPEC $M^{\prime }$ will get a bigger probability. The properties of semantic generalization can be reflected by their transition functions and the languages accepted.

An important difference between DPEC and NPEC is that a DPEC is a traditional probabilistic automaton but an NPEC $M^{\prime }$ is not a traditional probabilistic automaton because it does not satisfy complete probability formula. In PA *M*, $\sum_{p\in Q}\delta \left(q,a,p\right)=1$. But in corresponding NPEC $M^{\prime }$, because $\delta^{\prime }\left(q,a,p\right)\geq 0$, when $a\in \Sigma^{\prime }$,
$$\sum_{p\in Q}\delta^{\prime }\left(q,a,p\right)\geq \sum_{p\in Q}\delta \left(q,a,p\right)=1.$$

If we want to define a NPEC $M^{\prime }$ as a kind of traditional probabilistic automaton, the transition function needs to be adjusted as follows.

**Definition** **10**(NPEC)**.** *A nondeterministic probabilistic automaton for semantic computing under equivalent concepts (NPEC) is a seven-tuple $M^{\prime }=\left(Q,\Sigma ,\delta ,\Sigma^{\prime },\delta^{\prime },q_{0},F\right)$, where*
$M=(Q,\Sigma ,\delta ,q_{0},F)$ is a PA,$\Sigma^{\prime }=\left\{a|a\in \pi ,\exists a^{e}\in \Sigma \right\}$, $N(a^{e})$ is the number of the equivalent concepts of a,*$\delta^{\prime }:Q\times {\left(\Sigma \cup \Sigma^{\prime }\right)}^{*}\to D\left(Q\right)$ is a generalized transition function of δ:*$$\delta^{\prime }\left(q,a\right)=\left\{\begin{array}{c} \delta (q,a),\;\;if\;\;a\in \Sigma ,\\ \frac{1}{N(a^{e})}\sum_{a^{e}\in \Sigma }\delta \left(q,a^{e}\right),\;\;if\;\;a\in \Sigma^{\prime }. \end{array}\right.$$

It is obvious that the properties of robustness and semantic generalization are still valid and by the above adjusted transition function, it is easy to get the following property.

**Theorem** **4**(Semantic computing)**.** *In a DPEC or NPEC $M^{\prime }=\left(Q,\Sigma ,\delta ,\Sigma^{\prime },\delta^{\prime },q_{0},F\right)$, for any $a\in \Sigma^{\prime }$, $\sum_{p\in Q}\delta^{\prime }\left(q,a\right)\left(p\right)=sim\left(a,a^{e}\right)=1$.*

**Proof.** In a DPEC or NPEC $M^{\prime }=\left(Q,\Sigma ,\delta ,\Sigma^{\prime },\delta^{\prime },q_{0},F\right)$,
for any a∈Σ, $\sum_{p\in Q}\delta^{\prime }\left(q,a\right)\left(p\right)=\sum_{p\in Q}\delta \left(q,a\right)\left(p\right)=1=sim\left(a,a^{e}\right)$,for any $a\in \Sigma^{\prime }$,
$$\begin{array}{c} \sum_{p\in Q}\delta^{\prime }\left(q,a\right)\left(p\right)\\ =\sum_{p\in Q}\frac{1}{N(a^{e})}\sum_{a^{e}\in \Sigma }\delta \left(q,a^{e}\right)\left(p\right)\\ =\frac{1}{N(a^{e})}\sum_{a^{e}\in \Sigma }\sum_{p\in Q}\delta \left(q,a^{e}\right)\left(p\right)\\ =\frac{1}{N(a^{e})}\sum_{a^{e}\in \Sigma }1\\ =\frac{1}{N(a^{e})}*N\left(a^{e}\right)\\ =1\\ =sim(a,a^{e}). \end{array}$$ □

The intuitive idea of this theorem is that the total probability of that a DPEC or NPEC $M^{\prime }$ will enter all the states with a similar input is the semantic similarity between generalized input and similar input. The main purpose of a DPEC (or NPEC) $M^{\prime }=\left(Q,\Sigma ,\delta ,\Sigma^{\prime },\delta^{\prime },q_{0},F\right)$ generalized from PA $M=(Q,\Sigma ,\delta ,q_{0},F)$ is semantic computing.

From the above theorems, we can get that all the generalized PA do not have the limitations coming from the fixed finite input alphabet. They explain semantic computing with computation theory, instead of implementations of semantic reasoning (e.g., ontology reasoning, rule reasoning, semantic query, and semantic search).

### 3.2. Probabilistic Automata under Related Concept

In the previous subsection, we suppose that for any input a∉Σ, there exists an $a^{e}\in \Sigma$. In fact, it is just an ideal case. As illustrated in bookshop example, when a PA $M=(Q,\Sigma ,\delta ,q_{0},F)$ is applied to an application, in order to handle unpredictable inputs, we rebuild it as a NPEC $M^{\prime }=\left(Q,\Sigma ,\delta ,\Sigma^{\prime },\delta^{\prime },q_{0},F\right)$. When an input a∉Σ transmitted from users, we check the original input alphabet Σ. If there exist a $a^{e}\in \Sigma$ such that $sim(a,a^{e})=1$, then we add *a* to $\Sigma^{\prime }$ and define $\delta^{\prime }\left(q,a\right)=\delta \left(q,a^{e}\right),q\in Q$. But it is too hard to find a $a^{e}\neq \varepsilon \in \Sigma$ such that $sim(a,a^{e})=1$ for every input *a*. In fact, most of the time, we can only find a $a^{*}\in \Sigma$ such that $0\leq sim(a,a^{*})\leq 1$. Therefore, in this subsection, we generalize DPEC and NPEC to the case $0\leq sim(a,a^{*})\leq 1$.

**Definition** **11.**
*Suppose *Σ* is an alphabet and π is the alphabet of all the possible symbols. For any b∈Σ,a∈π, if $sim\left(a,b\right)=max_{b^{\prime }\in \Sigma }\left\{sim\left(a,b^{\prime }\right)\right\}>0$, then b is called the most similar concepts of a in *Σ*, denoted as $a^{*}=b$. If a∈Σ then $a^{*}=a^{e}=a$. If $max_{b^{\prime }\in \Sigma }\left\{sim\left(a,b^{\prime }\right)\right\}=0$, then $a^{*}=\varepsilon$. Notice that $a^{*}$ is not unique.*


With the most similar concepts, we generalize DPEC and NPEC to the universal case as follows.

**Definition** **12**(DPRC)**.** *A deterministic probabilistic automaton for semantic computing under related concept (DPRC) is a seven-tuple $M^{\prime }=\left(Q,\Sigma ,\delta ,\Sigma^{\prime },\delta^{\prime },q_{0},F\right)$, where*
$M=(Q,\Sigma ,\delta ,q_{0},F)$ is a PA,$\Sigma^{\prime }=\left\{a|a\in \pi ,\exists !a^{*}\in \Sigma \right\}$,*$\delta^{\prime }:Q\times \left(\Sigma \cup \Sigma^{\prime }\right)\to D\left(Q\right)$ is a generalized transition function of δ:*$$\delta^{\prime }\left(q,a\right)=\left\{\begin{array}{c} \delta (q,a),\;\;if\;\;a\in \Sigma ,\\ \delta \left(q,a^{*}\right)\cdot sim\left(a,a^{*}\right),\;\;if\;\;a\in \Sigma^{\prime }. \end{array}\right.$$

**Definition** **13**(NPRC)**.** *A nondeterministic probabilistic automaton for semantic computing under related concept (NPRC) is a seven-tuple $M^{\prime }=\left(Q,\Sigma ,\delta ,\Sigma^{\prime },\delta^{\prime },q_{0},F\right)$, where*
$M=(Q,\Sigma ,\delta ,q_{0},F)$ is a PA,$\Sigma^{\prime }=\left\{a|a\in \pi ,\exists a^{*}\in \Sigma \right\}$,*$\delta^{\prime }:Q\times \left(\Sigma \cup \Sigma^{\prime }\right)\to D\left(Q\right)$ is a generalized transition function of δ:*$$\delta^{\prime }\left(q,a\right)=\left\{\begin{array}{c} \delta (q,a),\;\;if\;\;a\in \Sigma ,\\ \sum_{a^{*}\in \Sigma }\delta \left(q,a^{*}\right),\;\;if\;\;\sum_{a^{*}\in \Sigma }\delta \left(q,a^{*}\right)\leq 1,a\in \Sigma^{\prime },\\ 1,\;\;if\;\;\sum_{a^{*}\in \Sigma }\delta \left(q,a^{*}\right)>1,a\in \Sigma^{\prime }. \end{array}\right.$$

Obviously, DPEC and NPEC are special cases of DPRC and NPRC, that is, $sim\left(a,a^{*}\right)=sim\left(a,a^{e}\right)=1$.

**Example** **3**(Command order)**.** *Recall the PA M in above Example 2. Here we consider the case that users may input “fit” instead of “right” or “true”. The term “fit” is not equivalent concept of “right” or “true”, but they are similar. Suppose sim(fit,right)=0.7,sim(fit,true)=0.8. Then we can only find a similar concept for “fit” instead of an equivalent concept. So DPEC and NPEC are not valid in this case. In fact, the probability of getting a similar input is bigger than an equivalent concept.*
*In order to deal with similar inputs, we rebuild PA M as a DPRC $M^{\prime }=\left(Q,\Sigma ,\delta ,\Sigma^{\prime },\delta^{\prime },q_{0},F\right)$ as follows. In this case, $\Sigma^{\prime }=\left\{fit\right\}$. Define $\delta^{\prime }\left(k,a,k^{\prime }\right)=\delta \left(k,a,k^{\prime }\right)$, for $k,k^{\prime }\in Q,a\in \Sigma$. $\delta^{\prime }$ is defined as:*
$$\begin{array}{c} \delta^{\prime }\left(q,fit,p\right)=sim\left(fit,true\right)\cdot \delta \left(q,true,p\right)=0.24,\\ \delta^{\prime }\left(q,fit,q\right)=sim\left(fit,true\right)\cdot \delta \left(q,true,q\right)=0.56,\\ \delta^{\prime }\left(p,fit,q\right)=sim\left(fit,true\right)\cdot \delta \left(p,true,q\right)=0.08,\\ \delta^{\prime }\left(p,fit,p\right)=sim\left(fit,true\right)\cdot \delta \left(p,true,p\right)=0.72. \end{array}$$

*With the above extended $\Sigma^{\prime }$ and $\delta^{\prime }$, when we suppose to get “true” and “right” from users, but users input “fit”, PA M is invalid. However, DPRC $M^{\prime }$ is still valid.*

*Suppose sim(fit,right)=sim(fit,true)=0.8, then PA M need to be rebuild as a NPRC $M^{\prime }=\left(Q,\Sigma ,\delta ,\Sigma^{\prime },\delta^{\prime },q_{0},F\right)$ as follows. $\Sigma^{\prime }=\left\{fit\right\}$. Define $\delta^{\prime }\left(k,a,k^{\prime }\right)=\delta \left(k,a,k^{\prime }\right)$, for $k,k^{\prime }\in Q,a\in \Sigma$. $\delta^{\prime }$ is defined as:*
$$\begin{array}{c} \delta^{\prime }\left(q,fit,p\right)=sim\left(fit,right\right)\cdot \delta \left(q,right,p\right)+sim\left(fit,true\right)\cdot \delta \left(q,true,p\right)=0.48,\\ \delta^{\prime }\left(q,fit,q\right)=min\left\{sim\left(fit,right\right)\cdot \delta \left(q,right,q\right)+sim\left(fit,true\right)\cdot \delta \left(q,true,q\right),1\right\}=1,\\ \delta^{\prime }\left(p,fit,q\right)=sim\left(fit,right\right)\cdot \delta \left(p,right,q\right)+sim\left(fit,true\right)\cdot \delta \left(p,true,q\right)=0.16,\\ \delta^{\prime }\left(p,fit,p\right)=min\left\{sim\left(fit,right\right)\cdot \delta \left(p,right,q\right)+sim\left(fit,true\right)\cdot \delta \left(p,true,p\right),1\right\}=1. \end{array}$$

*Spelling errors can also be took as similar inputs analogously.*


The transition function of a DPRC or NPRC $M^{\prime }$ can also be generalized to a function $\delta^{\prime }:Q\times {\left(\Sigma \cup \Sigma^{\prime }\right)}^{*}\to D\left(Q\right)$ and for convenience, we still denote it as $\delta^{\prime }$:$$\delta^{\prime }\left(q,A\right)=\left\{\begin{array}{c} \delta^{\prime }\left(q,\varepsilon \right)=1/q,\;\;if\;\;A=\varepsilon ,q\in Q,\\ \delta^{\prime }\left(q,a_{1}a_{2}\right)=\sum_{p\in Q}\delta^{\prime }\left(q,a_{1}\right)\left(p\right)\cdot \delta^{\prime }\left(p,a_{2}\right),\;\;if\;\;A=a_{1}a_{2}\;\;and\;\;a_{1}\in {\left(\Sigma \cup \Sigma^{\prime }\right)}^{*},\\ a_{2}\in \Sigma \cup \Sigma^{\prime },q\in Q. \end{array}\right.$$

By this generalized transition function, a DPRC (or NPRC) *M* inherits the robustness of DPEC (or NPEC).

**Theorem** **5**(Robustness)**.** *Suppose that $M^{\prime }$ is a DPRC or NPRC, then $M^{\prime }$ can take any string of symbols $A\in \pi^{*}$ as an input, i.e., the transition function of a DPRC or NPRC $M^{\prime }$ can be generalized to a function $\delta^{\prime }:Q\times \pi^{*}\to D\left(Q\right)$.*

**Proof.** The proof is similar to proof of robustness of DPEC and NPEC. □

**Definition** **14.**
*The language accepted by a DPRC or NPRC $M^{\prime }$ is defined as a function $L_{M^{\prime }}:\pi^{*}\to \left[0,1\right]$: for any $x=a_{1}a_{2}\cdots a_{n}\in \pi^{*}$,*
$$L_{M^{\prime }}\left(x\right)=\sum_{q\in F,q_{i}\in Q}\left\{\delta^{\prime }\left(q_{0},a_{1},q_{1}\right)\cdot \delta^{\prime }\left(q_{1},a_{2},q_{2}\right)\cdots \delta^{\prime }\left(q_{n-1},a_{n},q\right)\right\},i=1,2\cdots n-1.$$


A DPRC $M^{\prime }$ also inherits the relationship of accepted languages between DPEC and PA.

**Theorem** **6**(Semantic generalization)**.** *Suppose that DPRC $M^{\prime }=\left(Q,\Sigma ,\delta ,\Sigma^{\prime },\delta^{\prime },q_{0},F\right)$ is a semantic generalization of PA $M=(Q,\Sigma ,\delta ,q_{0},F)$ and the languages accepted by them are functions $L_{M^{\prime }}$ and $L_{M}$ respectively. The languages accepted by them have the following properties:*
*1*.for any $x\in \Sigma^{*}$, $L_{M^{\prime }}\left(x\right)=L_{M}\left(x\right)$,*2*.for any $x\in \pi^{*}$, there exist $x^{\prime }\in \Sigma^{*}$ such that $L_{M^{\prime }}\left(x\right)\leq L_{M^{\prime }}\left(x^{\prime }\right)=L_{M}\left(x^{\prime }\right)$.

**Proof.** if x=ε or $x\in \Sigma^{*}$, the proof is similar to the proof of semantic generalization of DPEC and NPEC. For any $x=a_{1}a_{2}\cdots a_{n}\in \pi /\Sigma^{*}$,
$$L_{M^{\prime }}\left(x\right)=\sum_{q\in F,q_{i}\in Q}\left\{\delta^{\prime }\left(q_{0},a_{1},q_{1}\right)\cdot \delta^{\prime }\left(q_{1},a_{2},q_{2}\right)\cdots \delta^{\prime }\left(q_{n-1},a_{n},q\right)\right\},i=1,2\cdots n-1.$$Since $sim(a,a^{*})\leq 1$, $\delta^{\prime }\left(q,a,p\right)=\delta \left(q,a^{*},p\right)\cdot sim\left(a,a^{*}\right)\leq \delta \left(q,a^{*},p\right)$, for any q,p∈Q. Let $x=a_{1}^{*}a_{2}^{*}\cdots a_{n}^{*}\in \Sigma^{*}$.
$$\begin{array}{c} L_{M^{\prime }}\left(x\right)\\ =\sum_{q\in F,q_{i}\in Q}\left\{\delta^{\prime }\left(q_{0},a_{1},q_{1}\right)\cdot \delta^{\prime }\left(q_{1},a_{2},q_{2}\right)\cdots \delta^{\prime }\left(q_{n-1},a_{n},q\right)\right\}\\ \leq \sum_{q\in F,q_{i}\in Q}\left\{\delta \left(q_{0},a_{1}^{*},q_{1}\right)\cdot \delta \left(q_{1},a_{2}^{*},q_{2}\right)\cdots \delta \left(q_{n-1},a_{n}^{*},q\right)\right\}\\ =L_{M^{\prime }}\left(x^{\prime }\right). \end{array}$$And since $x=a_{1}^{*}a_{2}^{*}\cdots a_{n}^{*}\in \Sigma^{*}$, by (1), we can get $L_{M^{\prime }}\left(x^{\prime }\right)=L_{M}\left(x^{\prime }\right).$ □

**Theorem** **7**(Semantic generalization)**.** *Suppose that DPRC $M^{\prime }=\left(Q,\Sigma ,\delta ,\Sigma^{\prime },\delta^{\prime },q_{0},F\right)$ and NPRC $M^{\prime \prime }=\left(Q,\Sigma ,\delta ,\Sigma^{\prime \prime },\delta^{\prime \prime },q_{0},F\right)$ are semantic generalizations of PA $M=(Q,\Sigma ,\delta ,q_{0},F)$ and the languages accepted by them are functions $L_{M^{\prime }}$, $L_{M^{\prime \prime }}$ and $L_{M}$ respectively. The languages accepted by them have the following properties:*
*1*.for any $x\in \Sigma^{*}$, $L_{M^{\prime \prime }}\left(x\right)=L_{M^{\prime }}\left(x\right)=L_{M}\left(x\right)$,*2*.for any $x\in \pi^{*}$, there exist $x^{\prime }\in \Sigma^{*}$ such that $L_{M^{\prime \prime }}\left(x\right)\geq L_{M^{\prime }}\left(x\right)\leq L_{M}\left(x^{\prime }\right)$.

**Proof.** if x=ε or $x\in \Sigma^{*}$, the proof is similar to the proof of semantic generalization of DPEC and NPEC. For any $x=a_{1}a_{2}\cdots a_{n}\in \pi /\Sigma^{*},p\in Q$,
$$\begin{array}{c} \delta^{\prime \prime }\left(q,a,p\right)\\ =min\{\sum_{a^{*}\in \Sigma }\delta \left(q,a^{*},p\right)\cdot sim\left(a,a^{*}\right),1\}\\ \geq \delta \left(q,a^{*},p\right)\cdot sim\left(a,a^{*}\right)\\ =\delta^{\prime }\left(q,a,p\right). \end{array}$$Therefore, $L_{M^{\prime \prime }}\left(x\right)\geq L_{M^{\prime }}\left(x^{\prime }\right)$. By the semantic generalization of DPRC, we can get that there exist $x^{\prime }\in \Sigma^{*}$ such that $L_{M^{\prime }}\left(x\right)\leq L_{M}\left(x^{\prime }\right)$. □

The above two theorems show the relationship between the language accepted by NPRC or DPRC $M^{\prime }$ and the corresponding PA *M*. Every element of *M* is included in $M^{\prime }$ and the languages accepted by *M* can be taken as a part of the languages accepted by $M^{\prime }$. Therefore, $M^{\prime }$ is a semantic generalization of *M*.

**Example** **4**(Bookshop)**.** *Let $M=(Q,\Sigma ,\delta ,q_{0},F)$ be a PA. Suppose that M is applied to an application of recommendation system for a book shop. When a customer inputs the name of a book and buys a book, PA M recommends a list of books, ordered by the probability of which book the customer will buy as well. Suppose that Q is the set of books, i.e.,*
$$\begin{array}{c} Q=\{q_{1}=\mathrm{Web}\;\mathrm{Ontology}\;\mathrm{Languages},q_{2}=\mathrm{Knowledge}\;\mathrm{Representation},q_{3}=\mathrm{Artificial}\;\mathrm{Intelligence},\\ q_{4}=\mathrm{Theory}\;\mathrm{of}\;\mathrm{Computation}\}. \end{array}$$
Σ *includes all combinations of the key words of every book’s name in Q which means Σ= {*Web, Ontology, Languages, Web Ontology, Web Ontology Languages, Knowledge, Representation, Knowledge Representation*⋯}. δ(q,a,p) means the probability of recommending book p to a customer after this customer has bought q with input a. The value of every δ(q,a,p) can be calculated by historical data which means recommendation system is based on customer’s purchasing history.*
*But as a user, when I have to input “Web Ontology Languages”, I prefer to input “WOL”—short for “Web Ontology Languages”—because it is more efficient to input an acronym. When I have to input “Knowledge Representation” or “Artificial Intelligence”, I also prefer to input “KR” or “AI” as well.*

*On the other hand, as a designer, the inputs transmitted from users are unpredictable. Users may input acronyms, synonyms or equivalent concepts in semantics, even some strange symbols coming from spelling mistakes. Therefore, when defining the input alphabet *Σ*, we need to collect synonyms or equivalent concepts of every element of *Σ*, as many as we can. But when *Σ* is very big, it is too hard to collect all the synonyms or equivalent concepts. Another worse case is that, as the development of globalization, the users may come from any country and input symbols with any language (English, Chinese, French, and so on). For robustness and universality, as a designer, it is too hard to define a PA M with the ability to handle all those cases, because PA is restricted by a fixed finite input alphabet *Σ* defined previously. But a NPRC is competent for this challenge.*
*Hence, we rebuild PA M as a NPRC $M^{\prime }=\left(Q,\Sigma ,\delta ,\Sigma^{\prime },\delta^{\prime },q_{0},F\right)$ as follows. Firstly, define $\Sigma^{\prime }=\;\{$*linguistics, represent*⋯} as the set of some possible inputs. For every $a\in \Sigma^{\prime }$, define $\delta^{\prime }\left(q,a,p\right)=\delta \left(q,a,p\right)\cdot sim\left(a,a^{*}\right)$. Notice that after we have defined the way to compute semantic similarity, we only need to collect familiar equivalents and synonyms as $\Sigma^{\prime }$. When an input $a\notin \Sigma \cup \Sigma^{\prime }$ transmitted from customers, we add a to $\Sigma^{\prime }$ and find $a^{*}$ based on function sim. Then define $\delta^{\prime }\left(q,a,p\right)=\delta \left(q,a^{*},p\right)\cdot sim\left(a,a^{*}\right),q,p\in Q$.**For instance, we consider a sample case. In PA $M=(Q,\Sigma ,\delta ,q_{0},F)$, Q={q=* Web Ontology Languages*, p = *Knowledge Representation*}. Σ={*Web, Ontology, Languages, Representation, Knowledge*}. $q_{0}=q,F=Q$. δ is defined as:*
$$\begin{array}{c} \delta (q,Web,q)=0.7,\delta (q,Ontology,q)=0.8,\delta (q,Languages,q)=0.9,\\ \delta (q,Representation,q)=0.1,\delta (q,Knowledge,q)=0.2,\\ \delta (q,Web,p)=0.3,\delta (q,Ontology,p)=0.2,\delta (q,Languages,p)=0.1,\\ \delta (q,Representation,p)=0.9,\delta (q,Knowledge,p)=0.8,\\ \delta (p,Web,q)=0.1,\delta (p,Ontology,q)=0.4,\delta (p,Languages,q)=0.5,\\ \delta (p,Representation,q)=0.7,\delta (p,Knowledge,q)=0.9,\\ \delta (p,Web,p)=0.9,\delta (p,Ontology,p)=0.6,\delta (p,Languages,p)=0.5,\\ \delta (p,Representation,p)=0.3,\delta (p,Knowledge,p)=0.1. \end{array}$$
*δ(q,Web,p)=0.3 means after a customer inputs key words* “Web” *searching for a book and finally buys book* “Web Ontology Languages”, *the probability of he will buy book* “Knowledge Representation” *is 0.3. Others are similar.**In order to deal with similar inputs, we rebuild PA M to a NPRC $M^{\prime }=\left(Q,\Sigma ,\delta ,\Sigma^{\prime },\delta^{\prime },q_{0},F\right)$ as follows. Suppose that we only consider the additional case that customers will input* “linguistics” *and* “represent” *searching for a book, that is, $\Sigma^{\prime }=\left\{linguistics,represent\right\}$ and $\delta^{\prime }\left(k,a,k^{\prime }\right)=\delta \left(k,a,k^{\prime }\right)$, for $k,k^{\prime }\in Q,a\in \Sigma$. Suppose*
sim(linguistics,Languages)=0.9,sim(represent,Representation)=0.7
*and for other case $a^{\prime }\in \Sigma^{\prime },a\in \Sigma ,sim\left(a,a^{\prime }\right)=0$. For $a\in \Sigma^{\prime }$, $\delta^{\prime }$ is defined as:*
$$\begin{array}{c} \delta^{\prime }\left(q,linguistics,p\right)=\delta \left(q,Languages,p\right)\cdot sim\left(linguistics,Languages\right)=0.9*0.1=0.09,\\ \delta^{\prime }\left(q,represent,p\right)=\delta \left(q,Representation,p\right)\cdot sim\left(represent,Representation\right)=0.7*0.9=0.63,\\ \delta^{\prime }\left(q,linguistics,q\right)=\delta \left(q,Languages,q\right)\cdots sim\left(linguistics,Languages\right)=0.9*0.9=0.81,\\ \delta^{\prime }\left(q,represent,q\right)=\delta \left(q,Representation,q\right)\cdot sim\left(represent,Representation\right)=0.7*0.1=0.07,\\ \delta^{\prime }\left(p,linguistics,q\right)=\delta \left(p,Languages,q\right)\cdot sim\left(linguistics,Languages\right)=0.9*0.5=0.45,\\ \delta^{\prime }\left(p,represent,q\right)=\delta \left(p,Representation,q\right)\cdot sim\left(represent,Representation\right)=0.7*0.7=0.47,\\ \delta^{\prime }\left(p,linguistics,p\right)=\delta \left(p,Languages,p\right)\cdot sim\left(linguistics,Languages\right)=0.9*0.5=0.45,\\ \delta^{\prime }\left(p,represent,p\right)=\delta \left(p,Representation,p\right)\cdot sim\left(represent,Representation\right)=0.7*0.3=0.21. \end{array}$$
*With the above extended $\Sigma^{\prime }$ and $\delta^{\prime }$, when an input “linguistics” that is not in *Σ* transmitted to automaton, PA M is invalid, but NPRC is valid.*


An NPRC can be modified as a traditional probabilistic automaton as follows.

**Definition** **15**(NPRC)**.** *A nondeterministic probabilistic automaton for semantic computing under related concept (NPRC) is a seven-tuple $M^{\prime }=\left(Q,\Sigma ,\delta ,\Sigma^{\prime },\delta^{\prime },q_{0},F\right)$, where*
$M=(Q,\Sigma ,\delta ,q_{0},F)$ is a PA,$\Sigma^{\prime }=\left\{a|a\in \pi ,\exists a^{*}\in \Sigma \right\}$,*$\delta^{\prime }:Q\times \left(\Sigma \cup \Sigma^{\prime }\right)\to D\left(Q\right)$ is a generalized transition function of δ:*$$\delta^{\prime }\left(q,a\right)=\left\{\begin{array}{c} \delta (q,a),\;\;if\;\;a\in \Sigma ,\\ \frac{1}{\sum_{a^{*}\in \Sigma }sim\left(a,a^{*}\right)}\sum_{a^{*}\in \Sigma }\delta \left(q,a^{*}\right)\cdot sim\left(a,a^{*}\right),\;\;if\;\;a\in \Sigma^{\prime }. \end{array}\right.$$

The proof of $\sum_{p\in Q}\delta^{\prime }\left(q,a\right)\left(p\right)=1$ is similar to NPEC. The key to proof this is that for different most similar concepts $a^{*},a^{**}$ of *a*,
$$sim\left(a,a^{*}\right)=sim\left(a,a^{**}\right)=max_{a^{\prime }\in \Sigma }\left\{sim\left(a,a^{\prime }\right)\right\}.$$

After the above modification of transition function, NPRC becomes a traditional probabilistic automaton. It is obvious that the property of robustness is still valid. But recall that the main purpose of this paper is to build computational models for semantic computing. Therefore, we prefer to modify it to the following forms.

**Definition** **16**(NPRC)**.** *A nondeterministic probabilistic automaton for semantic computing under related concept (NPRC) is a seven-tuple $M^{\prime }=\left(Q,\Sigma ,\delta ,\Sigma^{\prime },\delta^{\prime },q_{0},F\right)$, where*
$M=(Q,\Sigma ,\delta ,q_{0},F)$ is a PA,$\Sigma^{\prime }=\left\{a|a\in \pi ,\exists a^{*}\in \Sigma \right\}$, $N(a^{*})$ is number of $a^{*}$ corresponding to a,*$\delta^{\prime }:Q\times \left(\Sigma \cup \Sigma^{\prime }\right)\to D\left(Q\right)$ is a generalized transition function of δ:*$$\delta^{\prime }\left(q,a\right)=\left\{\begin{array}{c} \delta (q,a),\;\;if\;\;a\in \Sigma ,\\ \frac{1}{N(a^{*})}\sum_{a^{*}\in \Sigma }\delta \left(q,a^{*}\right)\cdot sim\left(a,a^{*}\right),\;\;if\;\;a\in \Sigma^{\prime }. \end{array}\right.$$

Based on the above form of NPRC, for any DPRC and NPRC, they have the following property.

**Theorem** **8**(Semantic computing)**.** *For any DPRC or NPRC $M^{\prime }=\left(Q,\Sigma ,\delta ,\Sigma^{\prime },\delta^{\prime },q_{0},F\right)$, for any $a\in \Sigma^{\prime }$, $\sum_{p\in Q}\delta^{\prime }\left(q,a\right)\left(p\right)=sim\left(a,a^{*}\right)$.*

**Proof.** For any DPRC $M^{\prime }$ and any $a\in \Sigma^{\prime }$,
$$\begin{array}{c} \sum_{p\in Q}\delta^{\prime }\left(q,a\right)\left(p\right)\\ =\sum_{p\in Q}\delta \left(q,a^{*}\right)\left(p\right)\cdot sim\left(a,a^{*}\right)\\ =sim\left(a,a^{*}\right)\sum_{p\in Q}\delta \left(q,a^{*}\right)\left(p\right)\\ =sim(a,a^{*})*1\\ =sim(a,a^{*}). \end{array}$$For any NPRC $M^{\prime }$ and any $a\in \Sigma^{\prime }$,
$$\begin{array}{c} \sum_{p\in Q}\delta^{\prime }\left(q,a\right)\left(p\right)\\ =\sum_{p\in Q}\frac{1}{N(a^{*})}\sum_{a^{*}\in \Sigma }\delta \left(q,a^{*}\right)\left(p\right)\cdot sim\left(a,a^{*}\right)\\ =\frac{1}{N(a^{*})}\sum_{a^{*}\in \Sigma }sim\left(a,a^{*}\right)\sum_{p\in Q}\delta \left(q,a^{*}\right)\left(p\right)\\ =\frac{1}{N(a^{*})}\sum_{a^{*}\in \Sigma }sim\left(a,a^{*}\right)\\ =\frac{1}{N(a^{*})}*N\left(a^{*}\right)*sim\left(a,a^{*}\right)\\ =sim(a,a^{*}). \end{array}$$ □

Therefore, in a DPRC or NPRC $M^{\prime }$, for any $a\in \Sigma^{\prime }$, $\sum_{p\in Q}\delta^{\prime }\left(q,a\right)\left(p\right)$ turn out to be the semantic similarity of two concepts: $sim(a,a^{*})$. It means the total probability of that a DPRC or NPRC will enter all the states with a similar input is the semantic similarity between generalized input and similar input.

In this subsection, we have defined different kinds of NPRC, because in different cases, we need different transition function. For an NPRC $M^{\prime }$ based on Definition 13, any generalized input is related to all similar inputs. But a NPRC $M^{\prime }$ based on Definition 15 is a traditional probabilistic automaton which satisfies the properties of traditional probabilistic automaton. For a NPRC $M^{\prime }$ based on Definition 16, it is a probabilistic automaton model for semantic computing.

In the above subsections, traditional probabilistic automata are generalized for semantic computing under equivalent and related concepts or words, respectively. Compared with traditional probabilistic automata, generalized probabilistic automata are more robust, because it is still valid when the input transmitted from users is not in the defined input alphabet Σ. It explains the semantic computing from computation theory. In the following section, we will show the robustness with an application for the weather forecast.

## 4. Application

The theory of computation has been applied to different areas such as social networks [23], clustering [24], and e-Services [25]. In this paper, we will build a machine for weather forecast [26,27,28,29] based on automata.

For a given location and time, the primary mission of weather forecast is to predict the conditions of the atmosphere. Human beings have been attempted to predict the weather informally for millennia, because we need weather warnings to protect life and property. Forecasts based on temperature and precipitation are essential to agriculture. We need to be prepared for floods and droughts in advance anyway. In our daily life, the weather forecast is an important factor for the plan of outdoor activities.

The understanding of atmospheric physics builds the foundation of modern numerical weather prediction [30]. After collecting quantitative data about the current state of the atmosphere at a given place, weather forecasts are made by using meteorology to predict how the atmosphere will change. The basic idea of numerical weather prediction is that: firstly, sample the state of the fluid at a given location and time. Then use the equations of fluid dynamics and thermodynamics to estimate the state of the fluid at some time in the future. In Reference [28], they propose that:
The weather station variables analyzed were air temperature, RH, 6-m (20 ft) wind speed, precipitation, cloud cover and solar radiation. Note that in keeping with standard fire weather terminology we will refer to the 6-m wind speed as the 20-ft wind speed throughout the rest of the paper.”

Therefore, in this section, we define a factor that impacts weather as a six-tuple
a=(airtemperature,RH,6-mwindspeed,precipitation,cloudcover,solarradiation).

A numerical representation of the above six-tuple is called a weather factor, which means that the change of weather state is based on the changing trend of—temperature, RH, 6-m wind speed, precipitation, cloud cover and solar radiation. We want to build a probabilistic automaton PA $M=(Q,\Sigma ,\delta ,q_{0},F)$ for weather forecast based on finite historical observation data. The intuitive idea is that *Q* is the set of weather states: sunny, cloudy, rainy, and so on. Σ is the set of weather factors. δ(q,a)(p) represents the probability of weather state changing to *p* from *q* with weather factor *a*.

The challenge is that the input alphabet of traditional probabilistic automata is a fixed finite set. But in the natural environment, the weather has been changing all the time. It is impossible and impractical to define every case of weather factors. We will analyze the problem of lacking data in Table 1 lately. One compromised solution is that we extract a finite set of weather factors from historical observation data, defined as Σ. When it comes to a new weather factor a∉Σ, we find a most similar weather factor $a^{*}\in \Sigma$ such that $sim\left(a,a^{*}\right)=max_{a^{\prime }\in \Sigma }\left\{sim\left(a,a^{\prime }\right)\right\}$. Then we take *a* as $a^{*}$. With this idea, we rebuild PA *M* as a DPRC $M^{\prime }=\left(Q,\Sigma ,\Sigma^{\prime },\delta ,\delta^{\prime },q_{0},F\right)$ based on historical observation data for weather forecast.

Firstly, we build a PA $M=(Q,\Sigma ,\delta ,q_{0},F)$. For convenience, the six factors of temperature, RH, 6-m wind speed, precipitation, cloud cover and solar radiation are normalized to unit interval I=[0,1] by normalize function:$$f\left(x\right)=\frac{x-min}{max-min},$$
where min and max represent the minimum and maximum value that take into account. For instance, if we only consider the weather temperature between −20 degree centigrade and 50 degree centigrade, then every temperature *t* is normalized by:$$f\left(t\right)=\frac{t+20}{70}.$$

Suppose that Σ is a finite set of weather factors extracted from historical observation data. Because of normalization, every a∈Σ is a tuple $a=\left(a_{1},a_{2}\cdots a_{6}\right),\left|a_{i}\right|\in I$ where $a_{i}$ corresponding to the change of six factors: temperature, RH, 6-m wind speed, precipitation, cloud cover and solar radiation respectively. The sign of $a_{i}$ represents the corresponding factor is increasing (+) or decreasing (−) and the absolute value $|a_{i}|$ describe the rate of change. For instance, if $a_{1}=-0.2$, then it means the temperature is decreasing and its rate is 0.2. $a_{2}=0.3$ means RH is increasing and its rate is 0.3. In fact, a∈Σ is the changing trend of weather states. δ is defined as $\delta \left(q,a\right)\left(p\right)=\frac{n(q,a,p)}{\left|\Sigma \right|}$ where |Σ| is the total of all weather factors in Σ and n(q,a,p) is the number of times that the weather in state *q* changes to state *p* with weather factor *a* in historical observation data. It means the probability of weather in state *q* change to state *p* with weather factor *a* is n(q,a,p)/|Σ|. Let F=Q and $q_{0}$ be the first observation record. Then PA *M* can show the probability of weather states changing from a state to another state based on historical observation data.

Suppose that we have been recorded weather factors for Σ every 10 min for 10 years. Roughly, the total number of records in Σ is $10*365*24*6=525,600\approx 5.3\times 10^{5}$. However, if the six elements in a weather factor a∈Σ is accurate to 0.01, every element will has 201 cases between −1.00 to 1.00. Roughly, the number of total possible cases of weather factors is ${\left(200\right)}^{6}\approx 6.4\times 10^{13}$. Hence, for a random observed weather factor *a*, the probability of a∈Σ is:$$P\left(a\in \Sigma \right)=\frac{5.3\times 10^{5}}{6.4\times 10^{13}}\approx 0.$$

Table 1 show the approximate probability of a∈Σ in different cases. For example, the data in fist line means that if we record weather factors every 10 minutes for 10 years, we can get approximate $5.3\times 10^{5}$ records and the probability of a new observed weather factor a∈Σ is near 0.

Table 1 shows that even when we collect data from records over 100 years, it is still not enough to build Σ for PA *M*. Therefore PA *M* is not competent for the weather forecast. Because when we observe and get the current weather factor *a*, the probability of a∈Σ is almost 0. The weather has been changing all the time. There are so many possible cases of weather factors, even though we have been recording every minute for 100 years, it is still far from enough. It seems impossible to make sure that every time the observed weather factor *a* will be in the set Σ. Once the current weather factor *a* is not in Σ, PA *M* cannot work.

In order to build an automaton for weather forecast, we need to redefine PA *M* as a DPRC $M^{\prime }=\left(Q,\Sigma ,\delta ,\Sigma^{\prime },\delta^{\prime },q_{0},F\right)$ where $\Sigma^{\prime }$ is the set of all possible weather factors, that is, $\Sigma^{\prime }=E^{6},E=\left[-1,1\right]$. Recall that as the robustness of DPRC, $\delta^{\prime }$ can be generalized to a function $\delta^{\prime }:Q\times \pi^{*}\to D\left(Q\right)$. In this application, $\delta^{\prime }$ is generalized to $\delta^{\prime }:Q\times {\left(E^{6}\right)}^{*}\to D\left(Q\right)$.

The semantic similarity of two weather factors a,b∈Σ is defined as $sim:E^{6}\times E^{6}\to E$ and here we use the most classical cosine function to compute similarity, that is,
$$sim\left(a,b\right)=cos\left(a,b\right)=\frac{a\cdot b}{\left|a|\cdot |b\right|}=\frac{\sum_{i=1}^{6}a_{i}b_{i}}{\sum_{i=1}^{6}a_{i}^{2}\sum_{i=1}^{6}b_{i}^{2}},$$
where $a=(a_{1},a_{2}\cdots a_{6})$ and $b=(b_{1},b_{2}\cdots b_{6})$. Transition function $\delta^{\prime }$ is defined as
$$\delta^{\prime }\left(q,a\right)\left(p\right)=\left\{\begin{array}{c} \delta (q,a)(p),\;\;for\;\;a\in \Sigma ,\\ \delta \left(q,a\right)\left(p\right)\cdot sim\left(a,a^{*}\right),\;\;for\;\;a\in E^{6}/\Sigma , \end{array}\right.$$
where $a^{*}$ satisfy $sim\left(a,a^{*}\right)=max_{a^{\prime }\in \Sigma }\left\{sim\left(a,a^{\prime }\right)\right\}$.

The intuitive idea is to suppose the current weather factor is *a* and the weather state is *p*. When we want to know the probability of current weather *q* changing to state *p* with weather factor *a*, we go back to check the records. By semantic similarity function sim(a,b) we find that $a^{*}\in \Sigma$ is the most similar weather factor to *a*. The probability of *q* changing to *p* with weather factor *a* can be obtained from PA *M* by $\delta \left(q,a^{*}\right)\left(p\right)$. Then we can conclude that the probability of current state *q* changing to state *p* is $\delta \left(q,a^{*}\right)\left(p\right)\cdot sim\left(a,a^{*}\right)$. Then, by DPRC $M^{\prime }$, we can get the probability that the current weather state may change to any weather states with any weather factor.

Table 2 show the comparison of the data that needed for PA *M* and DPRC $M^{\prime }$. Suppose that we only have the data recorded every minute in last 10 years. Table 1 show the comparison of the data that needed for PA *M* and DPRC $M^{\prime }$. PA *M* needs $6.4\times 10^{13}$ to build Σ. The data we have is far more enough and it is impossible to get so many data. If we must get it, the cost will be out of control. But DPRC $M^{\prime }$ dose not has this problem. The data in the last 10 years is enough.

Another challenge is the precision of implements used to measure the weather factors. DPRC $M^{\prime }$ can only deal with precise information. Suppose that when measuring a weather factor, the result obtained from implements is *a*, but because of the precision of the implements, in fact the real weather factor may be $a+k_{a}$ ($k_{a}$ is deviation of *a*) as well. In this case, the weather factor is imprecise information which cannot be dealt with by DPRC $M^{\prime }$ but it can be described by distribution. Suppose $W_{a}=0.9/a+0.1/\left(a+k_{a}\right)$ means that when the result got from implements is *a*, the real weather factor may be *a* with the probability 0.9 and may be $a+k_{a}$ with the probability 0.1 caused by the precision of implements. In this case, we need to generalize above DPRC $M^{\prime }$ as a model $M^{\prime \prime }=\left(Q,\Sigma_{w},\delta ,\Sigma_{w}^{\prime },\delta^{\prime \prime },q_{0},F\right)$ where $\Sigma_{w}^{\prime }=\left\{W_{a}=0.9/a+0.1/\left(a+k_{a}\right)|a\in \Sigma^{\prime }\right\}$. $\delta^{\prime \prime }:Q\times \Sigma_{w}^{\prime }\to D\left(Q\right)$ is defined as:$$\begin{array}{c} \delta^{\prime \prime }\left(q,W\right)\left(p\right)\\ =\sum_{a\in \Sigma }\delta^{\prime }\left(q,a\right)\left(p\right)W\left(a\right)\\ =\sum_{a\in \Sigma }\delta^{\prime }\left(q,a\right)\left(p\right)W_{a}\left(a\right)\\ =0.9*\delta^{\prime }\left(q,a\right)\left(p\right)+0.1*\delta^{\prime }\left(q,a_{k}\right)\left(p\right)\\ =0.9*\delta \left(q,a\right)\left(p\right)*sim\left(a,a^{*}\right)+0.1*\delta \left(q,a_{k}\right)\left(p\right)*sim\left(a,a_{k}^{*}\right). \end{array}$$

Therefore, even though PA *M* is incompetent to weather forecast, but DPRC $M^{\prime }$ is competent. The critical difference is that PA *M* is restricted in a fixed finite input alphabet Σ, but DPRC $M^{\prime }$ can take any weather factor as input because of robustness. Even the input is not in the fixed input alphabet Σ defined previously, DPRC can take it as a similar case and give a similar result based on semantic similarity.

## 5. Conclusions

Traditional automata will be invalid when an undefined input is transmitted from users in applications. In this paper, we generalize probabilistic automata for semantic computing. The generalized automata are more robust, which can take any input as legal input. The semantic generalization of our new probabilistic automata can be reflected from the languages accepted by them, i.e., the languages accepted by a traditional probabilistic automaton are a part of the languages accepted by the corresponding probabilistic automaton under semantic similarity. As shown in the application of weather forecast, when a new weather factor is observed, traditional probabilistic automata become invalid, because the new weather factor hardly belongs to the input alphabet. For robustness, we rebuild traditional probabilistic automata based on semantic similarity. Then when a probabilistic automaton gets an undefined input transmitted from users, despite it is not in the original input alphabet, the automaton is still valid with this undefined input.

Furthermore, probabilistic automata under semantic similarity provide a different understanding of “semantic computing” from a computation theory perspective. In most traditional semantic computing models, computational implementations of semantic reasoning are based on ontology reasoning, rule reasoning, semantic query, and semantic search, and so on. But they are not from the perspective of the formal theory of computation. The semantic computing property of our new probabilistic automata bridges the gap between semantic computing and computation theory. They provide a solid foundation for semantic computing. In our future work, we are planning to generalize the formal models for semantic computing with timed automaton. We are also trying to apply the formal models for semantic computing to deal with the problems such as natural language processing and semantic search.

## Figures and Tables

**Table 1 entropy-21-00903-t001:** The approximate probability of a new observed weather factor a∈Σ in different cases.

Interval (Year)	Frequency (1/min)	Amount	Probability (Approximate)
10	10	$5.3\times 10^{5}$	0
10	1	$5.3\times 10^{6}$	0
50	10	$2.6\times 10^{6}$	0
50	1	$2.6\times 10^{7}$	0
100	10	$5.3\times 10^{6}$	0
100	1	$5.3\times 10^{7}$	0

**Table 2 entropy-21-00903-t002:** Comparison of the data that needed for PA *M* and DPRC $M^{\prime }$.

Automaton	How Many Data We Have	If the Data Is Enough	How Many Data Are Needed	Cost
PA *M*	$5.3\times 10^{6}$	no	$6.4\times 10^{13}$	out of control
DPRC $M^{\prime }$	$5.3\times 10^{6}$	yes	$5.3\times 10^{6}$	under control

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
