# Peer review of "A Formal Model for Semantic Computing Based on Generalized Probabilistic Automata"

_entropy, 2019, doi:10.3390/e21090903_

Round 1

Reviewer 1 Report

The authors aim to build a formal model for semantic computing based on automata. This work is an extension of their previous paper [11], in which a finite state automata is replaced by its probabilistic version. Generally speaking, the paper is easy to follow. But I have a concern on the application or experiments.

It seems to me authors only mention that their new model can be applied for weather forecasting. Unfortunately, I did not see any figures or tables to demonstrate the experimental results? The comparison is also missing. I expect a more rigorous treatment to Section 4.  

Author Response

Reviewer#1, Concern # 1: It seems to me authors only mention that their new model can be applied for weather forecasting. Unfortunately, I did not see any figures or tables to demonstrate the experimental results? The comparison is also missing. I expect a more rigorous treatment to Section 4. 

Author response:  Thank you for this suggestion.

(1):We provide 4 examples (about command order and book shop) and an application (about weather forecast) to show the advantages of our new models.

(2):In Section 4 Application, we provide more data and tables (Table 1 and Table 2) to show how our new models overcome the limitations of traditional models.

(3):We compare them in Table 2.

Reviewer 2 Report

Review of the paper entitled "A formal model for semantic computing based on generalised probabilistic automata", by Guangjian Huang, Shahbaz Hassan Wasti, Lina Wei and Yuncheng Jiang.

Summary:
********

This paper investigates probabilistic automata in the context of semantic computing. It provides a way to extend probabilistic automata (PA) in such a way to handle the possibility of processing input symbols that are not in their original alphabet. The general idea is that, for any input symbol lying outside an original alphabet, its processing is defined in terms of one or many symbols that are equivalent or similar to it, but inside an original alphabet. In other words, the processing of a priori "unconstrained" input symbols is defined in terms of "legal" inputs that are either equivalent or similar two them.

Towards this purpose, the concepts of "deterministic (non-deterministic) probabilistic automaton for semantic computing under equivalent (related) concepts" (DPEC, NPEC, DPRC, NPRC) are defined as generalisations of deterministic (non-deterministic) PA, in the sense described above. It is shown that the transition functions of the new PA can be generalised to any possible input symbols (robustness theorems). Also, the relationships between the languages accepted by these new PA and their underlying classical PA are given (semantic generalisation theorems). Moreover, the processing of unconstrained symbols is expressed in terms of their similarities with legal inputs (semantic computing theorems). In short, the languages accepted by a traditional PA are a part of the languages accepted by their generalised counterparts.

Finally, an application of this theoretical framework is provided in context of weather forecasting.

Appreciation:
**********

Overall, I have a very mixed opinion about the paper both in terms of .

First of all, I agree with the authors about the gap between classical automata theory and semantic computing, and in this respect, I consider their "bridging approach" legitimate and interesting.

In classical theoretical computer science, the common approach is to consider abstract machines working over finite alphabets - and in general, over the binary alphabet $\Sigma = {0,1}$. The reason for this is that any finite alphabet can be encoded in binary (for instance an alphabet with 4 symbols can be encoded by the 4 binary strings 00, 01, 10, 11). Note that even if the number of symbols (generally called "letters") is finite, the number of possible inputs (generally called "words"), which consist of sequences of letters, is by contrast infinite (for instance, over the alphabet {0,1}, one can generate the infinitely many words 0, 1, 00, 01, 10, 11, 000, 001, 011,...).

In this context, probabilistic automata represent a whole active field of research, in both contexts of finite and infinite words (i.e., finite or infinite input streams). The topic generally involves deep theoretical results in computability theory, formal language theory, complexity theory, etc. Among plenty of other works, see for instance:

- Chapter 2 of Marielle Stoelinga. Alea jacta est: verification of probabilistic, real-time and parametric systems. PhD thesis, University of Nijmegen, the Netherlands, April 2002. Available at https://wwwhome.ewi.utwente.nl/~marielle/thesis.html

- Hugo Gimbert, Youssouf Oualhadj. Probabilistic Automata on Finite words: Decidable and Undecidable Problems. ICALP 2010, Jul 2010, Bordeaux, France. pp. 527-538.

- Chatterjee K., Henzinger T.A. (2010) Probabilistic Automata on Infinite Words: Decidability and Undecidability Results. In: Bouajjani A., Chin WN. (eds) Automated Technology for Verification and Analysis. ATVA 2010. Lecture Notes in Computer Science, vol 6252. Springer, Berlin, Heidelberg.

On the other hand, semantic computing seems to be more focused on applications (I'm not an expert). In this context, the possible symbols processed by the abstract machines need to be directly part of the input alphabet, rather than being encoded by words over a binary alphabet. In this context, the question of extending the input alphabet in order to grasp new input symbols, raised in this study, arises.

According to these considerations, I mention once again that I consider the "bridging approach" of the authors as legitimate and interesting. But according to the references in the paper, it seems to me that the authors have only little knowledge about "classical" probabilistic automata (in theoretical computer science). And I have to confess that I found the proposed results rather straightforward, far less deep than what I'm used to read in classical automata theory. Nevertheless, this does not mean that they do not deserve publication. Rather, I think that the main interest of the proposed approach resides in its applicative rather than theoretical extent. Therefore, I would encourage the author to clean the theoretical part (based on the comments below) and significantly extend, develop and improve Section 4 into a more convincing application case.

In terms of form, unfortunately, I found he paper not very well written. Below, I point out several typos, sentences, mathematical expressions, etc. to be corrected and/or improved. Also, I found the tone employed by the authors sometimes a little bit presumptuous (even if it is probably not their intention). In particular, the authors consistently talk about "rebuilding" a theory, which might sounds a bit grandiose: "we rebuild traditional probabilistic automata for semantic computing." I also found the style fairly repetitive, in particular in the Introduction and in the examples.

To summarise, I have a mixed opinion about the paper. In order to improve it, I would suggest to:
- clean the theoretical part;
- significantly develop the applicative part;
- significantly improve and the style of writing.
For these reasons, I would opt for a major revision.

Typos, maths, and other mistakes:
****************************

Line 1: the term "computing" by "semantic computing" -> the term "semantic computing" or simply "semantic computing" refers to...

Line 4: It examples semantic computing from the computation theory perspective. -> This approach provides a computer theoretical approach to semantic computing.

Lines 7-9: Even though probabilistic automata can handle unprecise knowledge, but the limitation of traditional probabilistic automata is that they are defined on a fixed finite input set. -> Even though traditional probabilistic automata can handle imprecise knowledge, their limitation resides in that they are defined on a fixed finite input set.

Lines 10-11: the following sentence is odd and repetitive, I'd suggest to remove it: "It explains semantic computing from the perspective of computation theory."

Lines 12-14: We provide an application about weather forecast system. Traditional probabilistic automata are incompetent for this forecast system because of its finite input set. But our new probabilistic automata can overcome this limitation. -> We provide an application to weather forecast, a domain for which traditionalise probabilistic automata are are not effective due to their finite input set. Our new probabilistic automata can overcome these limitations.

From this point onwards, I'll be less precise...

Line 14: the term "computing" by "semantic computing" means -> "semantic computing" refers to

Line 38: The key to do this is the methods in semantic computing. -> wrong sentence

Lines 43-45: We will not get into details, because the main work of this paper is the formal models for semantic computing. -> inappropriate sentence

Lines 54-55: Moreover, the rebuilt automata explain semantic computing from the perspective of computation theory. -> Moreover, our proposed automata provide a computer theoretical approach to semantic computing.

Line 62: We denote $f$ as the notation in [21]: -> A probability distribution $f \in D(U)$ is denoted as follows [21]:

Line 66: important and throughout the whole paper: in computability theory, $\Sigma$ is generally called the "alphabet" - and not the set of alpahbets! An alphabet is already a set whose elements are called letters or symbols. For instance, the latin alphabet is {a, b, c, d, ...}. So what you have a "finite alphabet" not a "finite set of alphabets".

Line 68: is $q_0 = \Sum_{\alpha_i} q_i$ really an element of $Q$? This point appears several times in the next definitions of the paper.

Line 70: is denoted as -> denotes

Lines 82-83: We apply the results of those research to our work directly, and we will not get into the details, because our main works are computational models for semantic computing. -> inappropriate and odd sentence

Line 102: Here again and everywhere in the sequel: $\Sigma$ is an alphabet, not a set of alphabets! $\pi$ is the set of all possible symbols, not alphabets ($\pi$ is not a set of sets).

Line 106: function $L : ...$ -> function $L_M : ...$
Also, in the definition of $L(M)$, when define you set of products \{ \delta(q_0,a_1,q_1) \cdot ... \}, the $q_i \in Q, i = 1,2,...,n-1$ should be inside the brackets, correct? Please check if I'm wrong... The same problem appears later in Definitions 7, proof of Theorems 2 and 3, Definition 14 and proof of Theorem 6.

Line 116: set of alphabets -> alphabet

Line 120: you should find a way to denote the several $a^e \in \Sigma$...

Line 134: In Definition 6, I'm not sure about the definition of $\delta'(q,a)$. Following your notations, the expression $\Sum_{a^e \in \Sigma} \delta(q, a^e)$ is a probability distribution. Hence, in order for the expression inside your min to make sense, 1 should be considered as the constant function 1 (we cannot compare functions and scalars). But following your subsequent example (line 137), I'm not sure that this is what you really mean... Please check. The same problem appears in Definition 9, Definition 13.

Line 141: I'd suggest putting this outside of Example 2

Lines 142-143: There is a lot of existing research on semantic computing. We can apply them to PA M directly. -> inappropriate sentence

Lines 149-150: repetition of "both forms of transition"

Line 152: again, $\pi$ is the set of all possible symbols , not alphabets

Lines 164-196: I found the presentation of Example 3 a bit repetitive.

Line 212: Because..., so we get -> Since..., we get
Also, in point (2), is $x = a_1a_2 \cdots a_n$ an element of $\pi^*$ or of $\Sigma'^*$? Please check.

Line 213: Because -> Since

Line 221: same remark: is $x = a_1a_2 \cdots a_n$ an element of $\pi^*$ or of $\Sigma'^*$?

Lines 225-226: then DPEC and NPEC $M'$ will get a bigger probability. I'm not sure, but isn't it only NPEC that gets a bigger probability (not DPEC)?

Line 250: In above subsection -> In the previous subsection

Line 258: $\pi$ is the set of all possible symbols

Line 277: The the -> The

Line 299: $L_{M''}$ -> $L_{M''}(x)$

Lines 386-387: Even if we can define an infinite set for every case of weather factors, the time complexity and space complexity will be enormous. -> imprecise sentence

Line 399: the number of. How many times -> the number of times

Line 426: input transmitted -> input is transmitted

Line 433: Because the new weather factor is not in the original set of inputs. -> incomplete sentence

Author Response

Reviewer#2, Concern # 1: I would encourage the author to clean the theoretical part (based on the comments below) and significantly extend, develop and improve Section 4 into a more convincing application case.

Author response:  Thank you for the kind introduction about the background of automata. It helps me understand automata more deeply. In Section 4, we provide more data and tables (Table 1 and Table 2) to show how our new models overcome the limitations of traditional models. We show that traditional model have a big problem to collect  enough data for input alphabet, but our new models is competent.

Reviewer#2, Concern # 2: Clean the theoretical part.

Author response:  Thank you for the introduction about the background about automata. We have checked the theoretical parts and made necessary changes.

Reviewer#2, Concern # 3: Significantly improve the style of writing.

Author response:  We are really grateful for your patience and kindness to check our paper throughout. We have checked the language issues and made some necessary changes.

Reviewer#2, Concern # 4: Important and throughout the whole paper: in computability theory, $\Sigma$ is generally called the "alphabet" - and not the set of alpahbets! An alphabet is already a set whose elements are called letters or symbols. For instance, the latin alphabet is {a, b, c, d, ...}. So what you have a "finite alphabet" not a "finite set of alphabets".

Author response:  Thank you for patience and kindness to introduce the knowledge about the alphabet of automata.  We have checked  the way we use the term  “alphabet” and made necessary changes.

Reviewer#2, Concern # 5: Is $q_0 = \Sum_{\alpha_i} q_i$ really an element of $Q$? This point appears several times in the next definitions of the paper.

Author response:  Yes, it is. In probabilistic automata, the start state is $q_i$ with  probability $a_i$.

Reviewer#2, Concern # 6:In the definition of $L(M)$, when define you set of products \{ \delta(q_0,a_1,q_1) \cdot ... \}, the $q_i \in Q, i = 1,2,...,n-1$ should be inside the brackets, correct? Please check if I'm wrong... The same problem appears later in Definitions 7, proof of Theorems 2 and 3, Definition 14 and proof of Theorem 6.

Author response:  Thank you for checking the whole paper so carefully.  We have corrected the mistakes in those definitions and theorems.

Reviewer#2, Concern # 7: You should find a way to denote the several $a^e \in \Sigma$.

Author response:  Thank you. We have defined the equivalent concept $a^e$ for concept $a$ in definition 4.

Reviewer#2, Concern #8: In Definition 6, I'm not sure about the definition of $\delta'(q,a)$. Following your notations, the expression $\Sum_{a^e \in \Sigma} \delta(q, a^e)$ is a probability distribution. Hence, in order for the expression inside your min to make sense, 1 should be considered as the constant function 1 (we cannot compare functions and scalars). But following your subsequent example (line 137), I'm not sure that this is what you really mean... Please check. The same problem appears in Definition 9, Definition 13.

Author response:  Thank you for point out this mistake. We have redefined the transition function in an appropriate way.

Round 2

Reviewer 1 Report

Authors addressed most of my concerns. But I still believe experiments can be improved for future work.